# Pathogenic Drug Resistant Fungi: A Review of Mitigation Strategies

**DOI:** 10.3390/ijms24021584

**Published:** 2023-01-13

**Authors:** Mary Garvey, Neil J. Rowan

**Affiliations:** 1Department of Life Science, Atlantic Technological University, F91 YW50 Sligo, Ireland; 2Centre for Precision Engineering, Materials and Manufacturing Research (PEM), Atlantic Technological University, F91 YW50 Sligo, Ireland; 3Bioscience Research Institute, Technical University Shannon Midlands Midwest, N37 HD68 Athlone, Ireland

**Keywords:** fungi, antifungal drug resistance, decontamination, disease prevention, one health, risk mitigation

## Abstract

Fungal pathogens cause significant human morbidity and mortality globally, where there is a propensity to infect vulnerable people such as the immunocompromised ones. There is increasing evidence of resistance to antifungal drugs, which has significant implications for cutaneous, invasive and bloodstream infections. The World Health Organization (WHO) published a priority list of fungal pathogens in October 2022, thus, highlighting that a crisis point has been reached where there is a pressing need to address the solutions. This review provides a timely insight into the challenges and implications on the topic of antifungal drug resistance along with discussing the effectiveness of established disease mitigation modalities and approaches. There is also a need to elucidate the cellular and molecular mechanisms of fungal resistance to inform effective solutions. The established fungal decontamination approaches are effective for medical device processing and sterilization, but the presence of pathogenic fungi in recalcitrant biofilms can lead to challenges, particularly during cleaning. Future design ideas for implantable and reusable medical devices should consider antifungal materials and appropriates for disinfection, and where it is relevant, sterilization. Preventing the growth of mycotoxin-producing fungi on foods through the use of appropriate end-to-end processes is advisable, as mycotoxins are recalcitrant and challenging to eliminate once they have formed.

## 1. Introduction

Fungi are eukaryotic microbial species that present in either yeast, mould or dimorphic forms. Yeasts are single celled, and they reproduce by budding, whereas fungi are multicellular with long filaments that are termed hyphae which grow via an apical extension. Fungi are abundant in the natural environment including water, soil and air, and they proliferate easily in warm and humid climates [1]. Indeed, fungi are the primary decomposers present in many ecosystems, releasing degradative enzymes for decomposing actions. Non-pathogenic endophyte fungal species are present in most forms of plant life between the plant cells, where they produce alkaloid toxins which act as insecticides and against other invertebrate animals and vertebrates [2]. Nycorrhizal fungi have a symbiotic relationship with plants, affecting nutrient and water uptake, while other species are plant pathogens, and they are associated with crop destruction, thereby impacting food security [3]. Fungal species are increasingly associated with morbidity and mortality, and they have become a significant public health risk [4]. Annually, fungal pathogens are the cause of approximately 13 million infections and 1.5 million deaths globally [5]. As they are traditionally associated with severe infections of immunocompromised persons, fungal infections are increasingly being associated with immunocompetent persons, with high mortality rates [1]. Species such as *Cryptococcus, Candida, Aspergillus*, and *Pneumocystis* are associated with immunocompromised persons, with the dimorphic fungi including *Histoplasma, Blastomyces, Coccidioides* and *Paracoccidioides*, which affect immunocompetent persons [6].

As the primary or opportunistic pathogens of humans, the resultant disease or mycosis can be superficial, such as infection of skin, hair, nail, mucosal surfaces, and allergic reactions, or invasive fungal infections (IFIs) that affect the internal organs, which are progressive and often fatal [7]. Defects in cell-mediated immunity typically result in a decrease in the activity of CD4+ lymphocytes in HIV patients, and this is the major risk factor for *Pneumocystis* pneumonia [7]. To be classified as an invasive fungal disease (IFD), tissue damage must be observed via a histopathological exam, with the causative agent isolated from clinical samples and cultured [8]. Advances in medical procedures and increasing occurrences of medical surgical procedures and therapeutic treatment protocols have increased the rate of identifying opportunistic infections during intensive treatments [9]. Furthermore, fungal infectious diseases can complicate chronic conditions or co-morbidities in patients with asthma, cirrhosis, diabetes, cystic fibrosis (CF), chronic obstructive pulmonary disease (COPD), cancer, and infectious diseases, including COVID-19 and tuberculosis (TB) [5]. Co-infections with viral pathogens are particularly problematic, and the fungal species *Aspergillus* and *Mucor* are associated with increased mortality in patients presenting with COVID-19 [10]. Fungal meningitis resulting from *Aspergillus fumigatus* has been associated with parenteral injections of corticosteroids, while *Sarocladium kiliense* was detected in fungaemia in oncology patients receiving IV antinausea medication [7]. Blood cancers such as leukemia render the patients at high risk of invasive fungal diseases due to their prolonged and severe neutropenia as a result of anticancer therapeutic treatment [11]. The high prevalence of morbidity and mortality are observed in transplant patients, cancer patients, patients in intensive care units (ICU), and HIV, influenza and COVID-19 patients [12]. The emergence of fungal pathogens is related to climate change, agricultural techniques, occupational hazard, forest erosion, human migration patterns, and soil dispersion, patient immunosuppression, improved disease recognition, and diagnostic tests [13]. Antifungal resistance, antifungal drug tolerance, and biofilm formation directly contribute to rising cases of fungal morbidity and mortality [4]. Antifungal resistance is an absence of a discernable toxic effect on treated fungal species; whereas, antifungal tolerance is the emergence of a partial growth after 24 h that can be seen in susceptibility testing, including at inhibitory drug concentrations [12]. Resistance to one or more of the four antifungal drug classes, polyene, azoles, allylamines and echinocandins, is frequently observed [14]. The chemical structure formulae of related antifungal drugs can be sourced from published reviews [15,16].

In October 2022, the World Health Organisation (WHO) released their fungal priority pathogen list (FPPL), detailing the medium, high, and critically important fungal pathogens [17], highlighting the extent of the public health risk associated with fungal infectious diseases and the antimicrobial resistance (AMR) of major pathogens (Table 1). This timely and important report has three main areas for action including strengthening laboratory capacity and surveillance, sustainable research, development, and innovation, and public health interventions [17]. The rapid diagnosis of fungal diseases is a key factor in the early prevention and control, where tissue cultures and biopsies are the gold standard for diagnosing IFDs [1]. Fungal infections remain frequently underdiagnosed, which results in variable outcomes for patients. This review aims to highlight the clinically important fungal pathogens in line with the WHO FPPL, antifungal resistance, and the importance of preventative and diagnostic procedures to protect public health.

## 2. Clinical Significance of WHO Pathogen List

Like other microbial species, fungi possess impressive genetic plasticity, allowing them to adapt to their environmental niche and rapidly display resistance to chemical insults and AMR [34]. Fungal traits including a short generation time, a broad range of natural habits, and a eukaryotic cell structure makes them extremely virulent, thereby increasing their pathogenicity [12]. Their eukaryotic cell greatly hinders the therapeutic treatments as it predisposes the humans to the toxic side effects of antifungal drugs due to a decrease in the number of selective drug targets [4]. Fungi are associated with cutaneous infections, invasive fungal infections, and nosocomial blood infections or fungemia. Sepsis from fungemia also contributes to high mortality rates, particularly where *Candida* is the causative agent of the disease [35]. Antifungal and biocidal resistance and a lack of biocompatible therapeutic options for IFIs contributes greatly to the disease prevalence, where MDR and pan drug resistance is common amongst many nosocomial species including *Candida albicans*, non-*albicans Candida* strains (NAC), *Cryptococcus, Aspergillus*, and numerous dermatophyte species [4]. As such, emerging IFDs are associated with the difficulty in treating the infections and high rates of mortality globally [7].

### 2.1. Cutaneous Infection

Cutaneous mycoses are superficial fungal infections of the skin, hair, or nails (onychomycosis), which are the most important causative agents of disease, including dermatophytes species (*Microsporum, Trichophyton,* and *Epidermophyton*), *Malassezia furfur,* and *Candida* (*albicans* and non-*albicans Candida*). The prevalence rate of cutaneous or superficial fungal infections (SFI) is approximately 20–25% globally [36]. Immunocompromised persons are high risk of cutaneous infections, where homeless persons are an often-overlooked high risk group due to malnutrition, lack of healthcare, injury, and co-morbidities [37]. Homeless patients are also high risk for cutaneous fungal infection related complications including cellulitis and osteomyelitis [37]. Importantly, deep cutaneous fungal infections (DCFIs) have high rates of morbidity and mortality, particularly amongst immunocompromised patients, with mortality rates of 4% to 10% in localized infections and ca. 83% to 94% in disseminated cases [38]. Clinically, the diagnosis of cutaneous etiological agents involves both mycological and histological findings [39].

Most of the dermatophyte species are zoonotic, and they are also transmitted via soil and from person to person with associated conditions including tinea capitis, tinea corporis, tinea pedis, tinea unguium, and tinea faciei [40]. Dermatophytes invade the stratum corneum of the epidermis, and they are not typically associated with invasive diseases [41]. The treatment of extensive or invasive dermatophytosis relies on systemic antifungal therapy including griseofulvin, terbinafine, ketoconazole, fluconazole, itraconazole, posaconazole, voriconazole, and ravuconazole antifungals [39]. The dermal commensals *Malassezia* species are the causative agents of pityriasis versicolor (PV), *Malassezia folliculitis*, and seborrheic dermatitis [36]. Topical antifungal creams including zinc pyrithione, ketoconazole, and terbinafine are the primary treatment for PV with oral itraconazole and fluconazole, which is prescribed for persistent cases [42]. *Candida* is associated with oral thrush, vaginal candidiasis, oropharyngeal candidiasis, cutaneous candidiasis, paronychia, and onychomycosis [36]. *Aspergillus* species, namely, *A. fumigatus* or *A. flavus*, with *A. terreus* are associated with cutaneous infections, as they are primary or secondary pathogens [41]. Primary *Aspergillus* disease is related to skin abrasion due to injury, surgery, organ transplant, or burn wounds with secondary diseases associated with an invasive disease of the lungs [41]. Importantly, *Candida* and *Aspergillus fumigatus* are both listed as critically important on the WHO FPPL. Atopic dermatitis (AD) is a prevalent inflammatory skin disease, where chronic conditions can be associated with microbial infection-inducing bacterial and viral species. Studies have observed that the fungal species *Malassezia*, *Candida,* and dermatophyte species can be associated with chronic conditions [43]. Mycetoma is a WHO-recognized neglected tropical disease that is caused by the fungi eumycetoma, and the disease manifests as subcutaneous chronic granulomatous progressively morbid inflammatory disease affecting the skin, subcutaneous tissue, deep structures, and bones, resulting in the destruction, deformity, loss of function, and it may lead to mortality [29]. The clinical symptoms of skin mycosis can vary across species, and cutaneous infections are often misdiagnosed as skin neoplasms or necrotizing lesions resulting from a lack of a suitable treatment [38].

### 2.2. Invasive Infection

*Aspergillus, Cryptococcus,* and *Candida* spp. are the main fungal species associated with invasive fungal infections of the lungs, brain, and bloodstream, respectively [12]. Disseminated infections are typically caused by *Blastomyces*, *Coccidioides*, *Paracoccidioides*, *Histoplasma,* and *Cryptococcus* spp. [44]. The pulmonary system (lungs) are the most common site of IFIs [45]. Triazole-resistant *A. fumigatus* and MDR yeast including *Candida glabrata* and *Candida auris* are of particular concern [46]. IFIs are separated from superficial mycoses due to the involvement of blood and other sterile body tissues or organs, and they are categorized as serious, deep, deep-seated, disseminated, and systemic fungal infections [47]. To cause an IFI in a patient, the fungi must have the ability to grow at or above 37 °C to reach internal tissues, the ability to lyse tissues and absorb their components, and they must be able to evade the host’s immune system [7]. Clinically invasive fungal diseases affect many organs and deep tissues, causing endocarditis, meningitis, and respiratory infections, and they are not often detected in blood cultures [46,48]. Furthermore, the insertion of venous catheters and intravascular devices and medical interventions allow for infections with nosocomial IFDs [7]. Cryptococcal meningitis caused by *Cryptococcus neoformans* or *Cryptococcus gattii* is common in HIV patients, where both of the species possess an innate resistance to fluconazole, where a combination therapy with flucytosine is implemented to improve the fungal clearance [12]. Additionally, ca. 7% of systemic *Candida* infections display reduced azole susceptibility [44]. For invasive aspergillosis, voriconazole is typically administered, and amphotericin B (AMPB) and the echinocandins also show anti-aspergillus activity, whereas the *Aspergillus* species possess a resistance to fluconazole [41]. The effective treatment of IFIs is also impacted by the lack of an accurate diagnosis. The diagnosis of IFIs is challenging, as the tests are slow, with limited sensitivity and specificity, and they are typically quite expensive [46]. IFI diagnosis consists of three elements: clinical symptoms (fever, a cough, dyspnea, chest pain, and hemoptysis), which are not always present, imaging results, and the detection of the causative agent [45]. The diagnosis of pulmonary invasive aspergillosis, for example, is achieved via a computed tomography (CT) scan of the chest in a patient with the appropriate risk factors to observe the nodules that are surrounded by a halo, which is a radiological feature [33]. Many A. *fumigatus* isolates are resistant to triazoles and possess pan-azole resistance. *A. niger,* for example, has resistance to oral itraconazole and isovuconazole drugs, with *Aspergillus. terreus* and *Asperguillus nidulans* possessing a resistance to AMPB [49]. The FDA suggests that AMPB is the safest antifungal agent for the treatment of systemic fungal infections, irrespective of its side effects, long half-life, and liver and kidney toxicity [44].

### 2.3. Bloodstream Infection

The number of bloodstream infections (BSIs) with fungal etiological agents has increased in recent years. *Candida* species are responsible for 90% of the fungal BSIs, and they result in late-onset sepsis etiologies amongst neonates [50]. *Candida* BSIs have a mortality rate of 30–40%, regardless of the therapeutic treatments [51]. Interestingly, studies describe higher mortality rates among countries and regions, where Latin American has a rate of ca. 60% compared to 20% in Spain [52]. Additionally, 80% of *Candida* BSIs occur in immunocompetent patients with nosocomial co-morbidities [7]. Studies have demonstrated the risk factors including diabetes, neoplasm, neutropenia, renal insufficiency, immunosuppression, cardiovascular disease, surgery, and age for fungal BSIs [51]. The incidence rates of BSIs with fungal pathogens are ca. 4.1% and ca. 0.69% in ICU patients in developing and developed countries, respectively [53], and this is directly related to the use of antifungal drugs, immunosuppression, steroids, placement of central venous catheters, and the low immunity of patients [51]. Echinomycin is recommended for fungal infections, fluconazole, however, is commonly used, particularly in developing countries, and it is associated with high rates of candidemia mortality due to azole resistance [54]. Non-*albicans Candida* species are more and more commonly associated with fungal BSIs, which are causative agents, *Candida parapsilosis* is associated with BSIs or candidemia in younger populations, with *C. glabrata* and *Candidatropicalis* being associated with elderly patient cohorts [51]. *C. tropicalis* is often associated with severe and fatal candidiasis, while *C. parapsilosis* is associated with lower mortality rates [52].

The widespread use of antifungal agents and increasing AMR has encouraged the emergence of non-*albicans Candida* BSIs, and studies have reported the use of echinocandins and azoles in the emergence of *C. parapsilosis* and *C. glabrata*, respectively [50]. Indeed, the multi-antifungal resistant, emerging, biofilm-forming *Candida auris* is listed as critically important by the WHO FPPL, and it is associated with fungal BSIs with high mortality rates [4]. Importantly, the health authorities recognize that *C. auris* is one of the most high-risk nosocomial pathogens due to its high transmissibility, AMR, and biocidal resistance [55]. The occurrence of MDR in fungal species has increased since 2017, particularly in *C. auris* and *C. glabrata*, which demonstrate resistance to the echinocandin drug class and fluconazole, which are the two first-line mono-therapeutic drugs for invasive candidiasis [44]. *Aspergillus* BSIs are not common, and they are typically associated with the dissemination of invasive lung aspergillosis or the infection of critically ill patients [56]. The initiation of a fungal therapy is linked to mortality rates; studies have demonstrated that the effective treatment of fungal BSIs after 12 h of withdrawing blood samples is linked to high mortality rates, while the initiation of therapy within a 12 h period of blood sampling is linked to lower mortality rates [57]. Furthermore, optimal dosing, dosing intervals, and the duration of the treatment are important factors in drug efficacy, reduced patient toxicity, and the prevention of fungal resistance [57].

## 3. Efficacy of Disinfection Strategies for Addressing Fungal Pathogens

### 3.1. Reusable Medical Devices and Surgical Instruments

Despite advances in medicine and innovations in many underpinning fields including disease prevention and control, the Spaulding classification system, which was originally proposed in 1957, remains widely used for defining the disinfection and sterilization of contaminated re-usable medical devices and surgical instruments [58]. Medical devices are a common source of hospital acquired infections (HAIs), and they have accounted for between 60% and 80% of all bloodstream-, urinary tract-, and pneumonia-related HAIs [59]. For example, at least 18 million gastrointestinal endoscopies are conducted each year in the United States [60]. Each of these procedures involves use of surgical instruments or medical devices that contact the patient’s sterile tissue or mucous membrane [61]. However, there is a marked lack of published information on the relevance of priority WHO fungal pathogens and the contamination of reusable medical devices in terms of transmission post-processing and sterilization. Notably, fungal infections cause over 1.5 million deaths per year, and a quarter million of these deaths are caused by the genus *Candida* [62]. The mortality rate of invasive candidiasis (infections by *Candida*) can be greater than 40% due to there being limited treatment options and increased antifungal resistance [4,62]. To mitigate the risk of HAIs, the current methods for the safe processing of medical devices still rely upon the guiding classification system of Dr E. H. Spaulding, which was originally conceived and published over 50 years ago [61].

Spaulding’s underpinning hypothesis was that healthcare facilities should apply appropriate disinfection and sterilization methods to process medical devices and surgical instruments based on the degree of the patients’ risk of acquiring an infection due to their use. Spaulding’s system divides all of the medical devices into three discrete categories based on the severity of the perceived risk to the patients of acquiring an infection from their use.

Critical use items: Where a device enters the sterile tissue and must be sterile, which is defined as being free from viable microorganisms [58]. Items contaminated with any microorganism (including fungal species) are referred to as high risk to the patients if they are contaminated and enter the sterile tissue or vascular system, and they have a high potential for causing disease transmission [61]. Such items should be sterile, such as by using steam sterilization where it is possible. The examples include surgical instruments. Given that many items contain heat-sensitive materials, other appropriate sterilization modalities should be applied, including vaporized hydrogen peroxide (VH_2_O_2_), VH_2_O_2_ gas plasma, and ethylene oxide gas [63]. The use of liquid chemical sterilants may also be considered appropriate, such as formulations based on glutaraldehyde (GTA), peracetic acid (PA), hydrogen peroxide (HP), or ortho-phthalaldehyde (OPA) [61,63].

Semi-critical use items: When a device only comes into contact with the intact mucus membranes or nonintact skin, it should also be subjected to sterilization, or if this is not feasible due to its sensitive material composition or complex design features, then a high-level disinfection (HLD) process must be deployed at a minimum level that would be expected to kill all of the microorganisms, except for the bacterial endospores [63]. The examples of semi-critical items including “respiratory therapy, anaesthesia equipment, some endoscopes, laryngoscope blades and handles, esophageal manometry probes, endocavitary probes, nasopharyngoscopes, prostate biopsy probes, infrared coagulation devices, anorectal manometry catheters, cystoscopies, and diaphragm fitting rings” [61]. Depending on the regional claim requirements, disinfectants should demonstrate a broad spectrum of antimicrobial activity, and typically, the ability to eliminate at least 10^6^ (or 6-logs) of the mycobacterial cells on the contaminated surfaces of the medical devices. For the fungi of concern, mycobacteria are typically deemed to exhibit greater resistance to high-quality disinfectants, thus, mycobacterial cells are recognized as being representative (or bio-indicators) of the HLD process efficacy. The examples of chemical disinfectants authorized in the USA for HLD use include biocides such as GTA HP, OPA, hypochlorite, and PA with HP [64].

Non-critical use items: Which includes devices that are in contact with intact skin (but not mucous membranes), requiring low-level-to-intermediate-level disinfection [64]. The skin contains intact integumentary layers, and as such, it provides a natural barrier to the microorganisms. There remains a risk to the skin as a result of cross-contamination from the devices, but this risk is considered to be low. These risks can be practically reduced by the combination of physical removal and disinfection [63]. The examples of non-critical use items include blood pressure cuffs, bed surfaces and rails, patient furniture, and so forth [61].

Figure 1 illustrates the microbial resistance profile to applied disinfection and sterilization modalities. It should be noted that the overall pattern of resistance to applied lethal technologies may vary depending on the modality. Microorganisms with higher resistance are widely used to challenge and test the effectiveness of disinfection and sterilization methods. Mycobacterial cells and *Bacillus* endospores have been used as indicators of HLD and sterilization, respectively [61]. Fungi exhibit greater biocidal resistance to enveloped viruses (such as HIV and SARS-COV-2) and to Gram-positive and -negative vegetative bacterial cells. Fungi present in vegetative- and spore-forming morphologies can be further differentiated based upon these morphologies with increasing exposure to these applied lethal stresses. For example, *Aspergillus* spores are more tolerant to higher doses of UV-irradiation due to the protective peak absorption of pigments at a similar UV-C wavelength to that of DNA (ca. from 250 to 260 nm) [65,66]. However, fungi are considered to be more susceptible to high-level disinfection (HLD) compared to similarly treated non-enveloped viruses (such as norovirus), mycobacterial cells, and parasitic oocysts (*Cryptosporidium* species), or cysts (Giardia species) [67,68,69].

Over the last few decades, there has been increasing amounts of genomic evidence of innate and adaptive microbial resistance to chemical disinfectant methods along with adaptive tolerance to environmental stresses. For example, this has been particularly evident in bacteria that are exposed to food processing (such as osmotic, acid, heat or UV-stressors) [70] or chemical biocide stresses [71], where the tolerance has been attributed to the expression of specific molecular determinants, ranging from protective stress protein synthesis to antimicrobial efflux pumps. The best published evidence argues that established HLD treatment and sterilization modalities effectively kill the free living fungi [72], however, the presence of fungi in recalcitrant biofilms may harbor these pathogens on the medical devices in the processing conditions used [62,73,74,75]. This area requires attention in future research. There is also a commensurate need to investigate the efficacy of HLD and sterilization in parallel with new medical design features, material compatibility, and cleaning regimes [76,77].

To reduce these risks of biofilm-mediated infections (including fungal) which are transmitted via contaminated medical devices, it is proposed that we should review the achievable and appropriate instructions of the manufacturers in the cleaning and processing of complex devices, where there can be as many as 100 steps to address by healthcare workers in the Sterile Services Department [61]. This brings the margin of error to near zero, which represents a higher risk to the patients. For example, it has been proposed that we should elevate the classification of high-risk flexible endoscopes (such as duodenoscopes) from semi-critical to critical use, which entails a transition to using low-temperature sterilization modalities instead of routinely using high-level disinfection, thus, increasing the margin of safety for endoscope processing. Gastrointestinal (GI) endoscopes can become highly contaminated during their use, where the internal long narrow lumen can contain between 7 to 10 log_10_ enteric microorganisms, and the microbial load of colon is ca. 9 to 12 log_10_/mL [61]. Outbreaks have been associated with medical device transmission where there were no reported links to“inadequate cleaning, inappropriate disinfection, and damaged endoscopes, or flaws in the design of endoscopes or automated endoscope re-processor” [61]. Often, these devices have also been highlighted as causative agents in outbreaks of multidrug-resistant organisms (MDROs), in which there were no obvious breaches in the endoscope reprocessing procedures [78,79,80,81,82,83,84,85]. However, the role of fungal infections arising from transmission by contaminated reusable medical devices through biofilms needs further research, as where there is a need to co-develop clinical diagnostics may be underappreciated.

### 3.2. Central Venous and Urinary Catheters

Modern technology has allowed us to use a wider and newer variety of medical devices. The combination of an increasingly aging population and a consistently growing number of inserted devices is likely to escalate the occurrence of infectious complications related to medical devices [86]. The number of indwelling medical devices is increasing, and an increasing proportion of device-related infections are being caused by *Candida* spp. *Candida* spp. produce biofilms on synthetic materials, which facilitates the adhesion of the organisms to the devices and renders them relatively refractory to medical therapy. The management of device-related *Candida* infections can be challenging. The removal of the infected device is generally needed to cure the *Candida* infections caused by the medical devices. However, since the pathogenesis of *Candida* bloodstream infection is complicated, more studies are necessary to determine the role of catheter exchange in patients with both gastrointestinal tract mucositis and indwelling catheters. Kojaic and coworkers [86] noted that *C. albicans* biofilm formation has three developmental phases: the adherence of yeast cells to the device’s surface (early phase), the formation of a matrix with dimorphic switching from yeast to hyphal forms (intermediate phase), and the increase in the amount of the matrix material, taking on a three-dimensional architecture (maturation phase). Fully mature *Candida* biofilms have a mixture of morphological forms, and they consist of a dense network of yeasts, hyphae, and pseudohyphae in a matrix of polysaccharides, carbohydrate, protein, and unknown components. The organisms in biofilms behave differently from freely suspended cells with respect to antimicrobial resistance. Both the bacteria and *Candida* cells within biofilms are markedly resistant to antimicrobial agents [86].

*C. auris* has become a global threat as it can colonize the skin, medical devices, and hospital environments, causing nosocomial outbreaks of blood and urinary tract infections worldwide [62]. *Candida auris* can spread among patients in hospitals, and it is intrinsically resistant to one or more classes of antifungals, which makes it particularly difficult to treat in healthcare settings. Comparative genomics has demonstrated that *C. auris* has expanded families of transporters and lipases, as well as mutations and copy number variants, in genes/enzymes linked to increased resistance and virulence [2]. Understanding the evolution of emerging fungal pathogens such as *C. auris* will be useful for the design of antifungal drugs and therapies for susceptible patients, potentially improving the clinical outcomes.

Piktel et al. [87] reported on the number of antimicrobial agents with the ability to prevent device-associated infections, and these have been proposed as biomaterials coatings. Alternative methods are constantly being developed using established antimicrobial agents. A large number of these applications involve the coating of medical device surfaces with metallic nanoparticles, such as zinc oxide (ZnO NPs)6, silver (Ag NPs)7, copper (Cu NPs), or titanium (TiO_2_ NPs), and the mechanism of protective effects of those nanomaterials includes mostly the disruption of the microbial membranes and the prevention of microbial proliferation of the surface of the device or implant. Slamborova et al. [88] combined silver, copper, and titanium dioxide nanoparticles to establish long-term, broad-spectrum antifungal and antibacterial coverage, while maintaining the appropriate mechanical properties of the coating itself. Piktel et al. [87] revealed that a relatively low dose of nanomaterials, i.e., ranging from 0.78 to 0.625 ng mL^−1^, should be considered as fungicidal and bactericidal, as has been demonstrated for *C. albicans*. Importantly, the minimum biofilm inhibitory concentrations (MBIC) were not significantly higher than the bactericidal ones were; for the majority of strains, the MBIC value was not greater than 0.625 ng mL^−1^. Owing to their unique physicochemical features and low cytotoxicity, gold nanoparticles (Au NPs) have been widely used in biological and biotechnological applications as biocidal agents, drug delivery systems, photosensitizer, and molecular diagnostic tools [8]. Piktel et al. [87] assessed the antimicrobial efficiency of non-spherical gold nanoparticles in the shapes of rods (AuR NPs), peanuts (AuP NPs), and stars (AuS NPs), as well as porous spherical-like nanoparticles (AuSph (70C) NPs), which exhibited potent antifungal effects, which contrasts those of previous reports including microgram concentrations (µg mL^−1^) of gold nanoparticles.

## 4. Knowledge Gap in Molecular and Cellular Mechanisms Underpinning Disinfection of Fungal Pathogens including AMR Post-Treatment Modalities

Advanced studies on cell survival following antimicrobial processes also are of interest [89]. As an example, Farrell et al. [90] highlighted the potential of addressing a single composite study to address the relationship between the use of pulsed UV light irradiation and the simultaneous occurrence of molecular and cellular damage in clinical strains of *C. albicans*. This is particularly relevant, and it showed that the occurrence of late apoptotic and necrotic cell phonotypes can be detected in real-time using specific representative biomarkers. This coincides with the occurrence of irreversible fungal cell death, which may potentially supplement or replace the lengthy standard culture-based methods where there was good agreement between these indirect biomarker and direct culture-base enumeration approaches. Notably, this constituted the first study to investigate the mechanisms of cell destruction caused by pulsed UV using a sequential and simultaneous microbial protein leakage assay and the lipid hydroperoxidation in the cell membranes, specific patterns of reactive oxygen species generation, and nuclear damage of treated microbial cells using a Comet assay, along with the detection of specific apoptotic and necrotic stages. Design, testing, and validating the real-time markers to demonstrate irreversible fungal death will prove the effectiveness of the disinfection modalities.

## 5. Need for Improved In Vitro and In Vivo Compatibility Tests for Medical Devices Encompassing Antifungal and Disinfection Efficacy

Researchers have noted that the limitations of in vitro and animal models of chronic device-related infections are important in the context of advancing the med-tech sector, with implications for pressing research and clinical practice [74]. Ramstedt et al. [75] evaluated the efficacy of antimicrobial and antifouling materials for a urinary tract medical device that also enabled the transmission of fungal infections. These authors addressed the challenges of antimicrobial material testing, including surface characterization, biocompatibility, cytotoxicity, in vitro and in vivo tests, microbial strain selection, and hydrodynamic conditions, from the perspective of complying to the complex pathology of device-associated urinary tract infections. Standard assays should be developed that enable us to make comparisons between the inter-laboratory generated results of industries and academia to perform harmonized assessments of the antimicrobial properties of urinary tract devices in a reliable way that includes improving in vitro and in vivo biocompatibility testing. Moreover, the high risk of infection and its associated costs clearly underlines the need to provide patients with devices with the lowest possible risk of infection, and it emphasizes the need for innovative products that reduce the incidence rate of device-associated UTIs. Although standards are available for guiding the development of new devices with respect to biocompatibility (ISO 10993) and material characterization, no such guidance exists for the development of antimicrobial devices [75].

## 6. Mycotoxins and Appropriate Decontamination Strategies

Mycotoxins are secondary metabolites of mold and fungi; they are generally toxic to living organisms. This term, by general consensus, is used almost exclusively for fungi associated with food products and animal feed, excluding the toxins produced by mushrooms. Mycotoxins are secondary metabolites with no apparent function in the normal metabolism of fungi [91]. They are produced mainly, although not exclusively, when the fungus reaches maturity [91]. They are molecules with structures which vary from simple heterocyclic rings with molecular weights of up to 50 Da to groups with 6–8 irregularly arranged heterocylic rings with a total molecular weight of >500 Da, and they do not show immunogenicity. Studies have revealed the existence of at least around 400 different mycotoxins [92]. Hundreds of mycotoxins have been identified thus far, with some of them, such as aflatoxins, ochratoxins, trichothecenes, zearalenone, fumonisins, and patulin, being considered agro-economically important [91]. Several factors contribute to the presence of mycotoxins in food, such as climatic conditions, pest infestation, and poor harvest and storage practices. Exposure to mycotoxins, which occurs mostly by ingestion, leads to various diseases, such as mycotoxicoses and mycoses, which may eventually result in death [91,93]. Mycotoxins can enter the human and animal food chains through direct or indirect contamination. The indirect contamination of foodstuffs and animal feed occurs when any ingredient has been previously contaminated with a toxigenic fungus, and even though the fungus has been eliminated during the process, the mycotoxins remain in the final product [92]. Direct contamination, on the other hand, occurs when the product, food, or feed becomes infected with a toxigenic fungus, resulting in the subsequent formation of mycotoxins [94]. Thus, more than a hundred mycotoxins are known, and most of them are produced by some of the species belonging to one of three fungi genera: *Aspergillus*, *Penicillium,* and/or *Fusarium* [95]. According to the available literature the “presence of the following mycotoxins in pollen has been investigated or proved with appropriate analytical methods and analysis: Aflatoxins (AFs), ochratoxins (OTs), fumonisins (FBs), zearalenone (ZEN), deoxynivalenol (DON), and its acetoxy derivative, T-2 toxin (T-2), HT-2 toxin, fusarenon-X, diacetoxyscirpenol, nivalenol, neosolaniol, roridin A, verrucarrin A, α-β-dehydrocurvularin, phomalactone,6-(1-propenyl)-3,4,5,6-tetrahydro-5-hydroxy-4H-pyran-2-one, 5-[1-(1hydroxibut-2-enyl)]-dihydrofuran-2-one and 5-[1-(1-hydroxibut-2-enyl)]-furan-2-one” [96].

The main aflatoxins that are known about are called B1, B2, G1, and G2 based on their fluorescence under ultraviolet light (B = Blue; G = Green) and their mobility during thin-layer chromatography [92]. They are mainly produced by *A. flavus and Aspergillus parasiticus*. It is known that only 50% of the strains of these species produce aflatoxins and that some of the aflatoxigenic isolates produce up to 106 μg/kg of aflatoxins [92]. Due to their capacity to bind with the DNA of cells, aflatoxins affect protein synthesis besides contributing to the occurrence of thymic aplasia (congenital absence of thymus and the parathyroids, with a consequent deficiency in the cell immunity, which is also known as DiGeorge’s syndrome) [92]. Aflatoxins have oncogenic and immunosuppressive properties, inducing infections in people who have been contaminated with these substances. Ochratoxin A is a metabolite of *Aspergillus ochraceus,* and it has a chemical structure similar to that of aflatoxins. It is associated with nephropathy in all of the animals that have been studied to date [97]. Besides being recognized as nephrotoxic, ochratoxin A, it also shows hepatoxic, immunosuppressive, teratogenic, and carcinogenic behaviors [92].

Fumonisins are produced by several species of the genus *Fusarium*, especially by *Fusarium verticillioides* (previously classified as *Fusarium moniliforme*), *Fusarium proliferatum and Fusarium nygamai*, besides *Alternaria alternata f.sp. lycopersici* [92]. The presence of fumonisins in corn grains has been associated with cases of oesophageal cancer [98]. Fumonisins are also responsible for leukoencephalomacia in equine species and rabbits [92,99], and hepatotoxic, carcinogenic, and apoptosis (programmed cell death) effects in the livers of rats [100]. Patulin is isolated from *Penicillium griseofluvum,* and albeit inconclusively, it is considered to be toxic to animals and plants [92]. Other mycotoxins of notoriety include ergot alkaloids and trichothecenes, which have been extensively reviewed (such as [91]).

The fungal contamination of different feed/food, including pollen will be more frequent as a result of the occurrence of intensive climatic changes [96]. The quality of pollen can be significantly influenced by the presence of toxigenic fungi. Since it has been proved that the absence of microbial contamination in pollen does not exclude the presence of mycotoxins, mycotoxicological analyses should also be included as a regular control measure, together with microbiological tests. Since aflatoxins and ochratoxins have been proven to be carcinogenic substances, their presence in pollen is extremely undesirable. Therefore, it is important to monitor the mold and mycotoxin levels in feed/food in order to avoid adverse health effects. The contamination of food and feed by mycotoxins represents a serious health problem for humans and animals, as well as a considerable economic obstacle in African, Asian, and Latin American countries, where the trade balance is based on the exportation of commodities. The recognition of problems caused by mycotoxins in foods destined for human and animal consumption is undoubtedly the first step toward the implementation of programs which enable the adoption of appropriate measures for the prevention and reduction of this problem [92]. The chemical structures of the principal mycotoxins can be found in many published reviews [91,92].

Thus, the consumption of mycotoxins-contaminated feed causes a plethora of harmful responses from acute toxicity to many persistent health disorders with lethal outcomes, such as mycotoxicosis when it is ingested by animals. Therefore, the main task for feed producers is to minimize the concentration of mycotoxins by applying different strategies that are aimed at minimizing the risk of the mycotoxin effects on animal and human health. However, once the mycotoxins enter the production chain, it is hard to eliminate or inactivate them [93]. Notably, mycotoxin-producing fungi are readily destroyed by moderate levels of disinfection. However, given the recalcitrant nature of mycotoxins, emphasis should be placed on ensuring appropriate end-to-end food production and management to prevent the growth of mycotoxin-producing organisms, such as cleaning the grains and removing the kernels that harbor molds. The use of feed additives or supplements that decrease the risk of animal exposure to mycotoxins can be viewed as a means of enhancing animal welfare. These feed supplements are referred to as the substances blended into feed (e.g., mineral clay, microorganism, and yeast cell wall), adsorbing or detoxifying the mycotoxins in the digestive tract of animals (biological detoxification) [93]. In general, mycotoxins are mainly stable compounds under thermal processing conditions used in feed and food [101]. However, the different thermal food and feed treatments that can have different impacts on the mycotoxins include extrusion, cooking, frying, baking, canning, crumbling, pelleting, roasting, flaking, and alkaline cooking. Among the thermal treatments, the utilization of high-temperature processes demonstrates the greatest potential for mycotoxin reduction [93]. Kabak [102] noted that the application of extrusion at a temperature that is higher than 150 °C has a significant impact on the reduction of zearalenone and fumonisins, while the same conditions led to the moderate reduction of aflatoxins and deoxynivalenol. Oxidizing agents such as ozone and hydrogen peroxide have been used to decontaminate mycotoxin-contaminated raw feed and compound feed [103]. The application of microorganisms or enzyme systems to contaminated feeds can detoxify the mycotoxins by metabolism or degradation in their gastrointestinal tract. This process is an irreversible and environmentally friendly method of detoxification, as it does not leave toxic residues or unwanted by-products [93]. However, the levels of particular mycotoxins in feeds have been reduced, but so far, no single technique has been established that is equally efficient against the broad variety of mycotoxins that can co-occur in various commodities [93]. Furthermore, the procedures of detoxication that appear to be efficient in vitro will not necessarily maintain their effectiveness in an in vivo test.

## 7. Conclusions

Fungal pathogens represent a serious public health risk, where AMR-incorporating biocidal resistance has proliferated the issue. The WHO has announced a fungal priority pathogen list, further highlighting the seriousness of the disease risk of these potentially life-threatening organisms. Antifungal resistance is further augmented by a lack of novel antifungal therapeutic options and associated biocompatibility issues, thereby limiting the medical applications. Without efficient control measures, the critically important WHO listed pathogens such as *C. auris* and *C. neoformans* will continue to result in unacceptably high rates of mortality. Additionally, the emergence of new species such as the non-*ablicans Candida* BSIs will increase, leading to the proliferation of AMR and increasing the death rates, particularly in immunocompromised persons. As with all of the infectious diseases, prevention is the optimal way to mitigate disease outbreak and transmission. The application of effective disinfection and sterilization regimes, particularly in hospital settings, is vitally important, where a focus on fungal biofilm formation on indwelling medical devices is important. Currently, there is an ongoing drug resistance crisis globally, where fungal AMR is often overlooked in terms of diagnosis and pathogen monitoring. In order to more accurately monitor and respond to the actual number of fungal-mediated infections that is underestimated, there is a need to improve fungal diagnostic and detection methods along with effective communication of same to clinicians. The widespread application of antimicrobial therapeutics without having conducted more investigative studies should not be applied. There is a pressing need to understand the cellular and molecular mechanistic relationship between device reprocessing and the inactivation of biofilm-forming fungi in order to mitigate device-related transmission. Semi-critical devices should be reviewed to reduce the risk to the patient, where there is an unreasonable number of cleaning and processing steps to satisfy the margin of safety in the healthcare setting. Preventing the growth of mycotoxin-producing fungi on foods through the performance of appropriate end-to-end processes is advisable, as mycotoxins are recalcitrant and challenging to eliminate once they have been formed. Adopting the OneHealth approach will support and enable solutions to address this complex societal challenge.

## Figures and Tables

**Figure 1 ijms-24-01584-f001:**
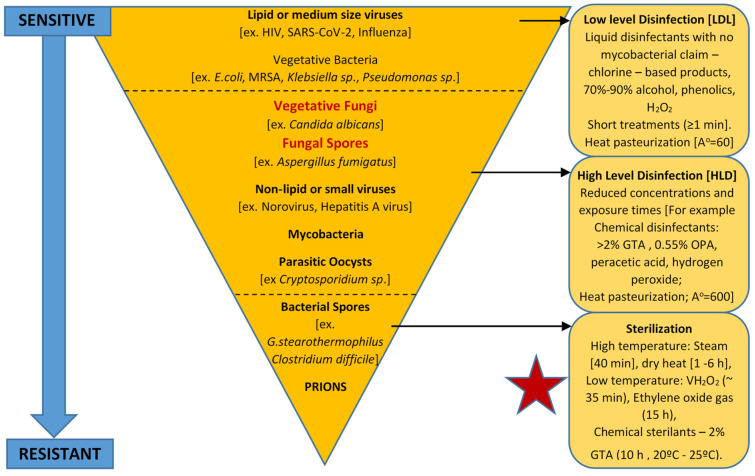
Pyramid of increasing microbial resistance to medical device processing and sterilization stressors.

**Table 1 ijms-24-01584-t001:** WHO priority pathogens, at risk patients, mortality rates, and AMR profile.

Priority	Pathogen	At Risk Patients	Mortality	AMR
Medium	*Scedosporium* spp.	Cystic fibrosis [18] and organ transplant recipients [19]	50% of organ transplant patients [19]	Itraconazole [12]
	*Lomentospora prolificans*	Cystic fibrosis patients [18], immunocompetent, and immune-suppressed people [13]	46.9% and 87.5% of patients with disseminated disease [13]	MDR, pan resistant [13]
	*Coccidioides* spp.	COVID-19-infected, pregnant and immunocompromised people [20]	70% of immunocompromised people [20]	Azoles
	*Pichia kudriavzeveii (Candida krusei)*	Neonates in ICU and immunocompromised people	30% [21]	Innately to fluconazole; Echinocandin resistance often evident [21]
	*Cryptococcus gattii*	AIDS/HIV patients [7]	10% [22]	Echinocandins [4]
	*Talaromyces marneffei*	AIDS/HIV patients [23]	13.3% [23]	N/A
	*Pneumocystis jirovecii*	AIDS/HIV and [5] *Cytomegalovirus* [7] patients	50% of immunocompromised persons [24]	Polyenes-amphotericin B (AMP B) and azoles [4]
	*Paracoccidioides* spp	AIDS/HIV and immunocompromised [25] patients	from 6.2% to 27% [25]	AMP B, ketoconazole, fluconazole, itraconazole, and sulfamethoxazole [26]
High	*Nakaseomyces glabrata (Candida glabrata)*	Renal failure and ICU patients [27]	54% [27]	Fluconazole, echinocandins [12], and echinocandins [4]
	*Histoplasma* spp.	AIDS/HIV patients and TB patients [7]	60% of AIDS/HIV patients	Azole
	*Eumycetoma-Madurella mycetomatis*	Barefoot walking populations of tropical and subtropical countries [28]	Very rare [29]	Prolong treatment needed which can lead to toxicity and surgery [29]
	*Mucorales*	Immunocompromised, COVID-19-infected [5], and diabetes mellitus patients [7]	80% [4]	Echinocandins [4]
	*Fusarium* spp.	Immunosuppressed, oncology, and organ transplant patients [30]	37%; disseminated fusariosis cases: 83% [30]	MDR (Garvey et al., 2022), intrinsically to azoles [30]
	*Candida tropicalis*	Nosocomial patients and oncology patients [31]	64–86% at 10–30 days [31]	Echinocandins [4]
	*Candida parapsilosis*	Nosocomial patients [32]	26% [32]	Echinocandins [4]
Critical	*Cryptococcus neoformans*	AIDS/HIV patients and TB patients [7]	20%, or 100% if untreated [4]	Fluconazole [12], flucytosine [4]
	*Candida auris*	Neonates, elderly, chronically ill, and on-therapy patients	70% [4]	Azoles, lower sensitivity to the polyene amphotericin B, MDR [12], echinocandins, and flucytosine [4]
	*Aspergillus fumigatus*	Cystic fibrosis [18], COVID-19-infected [5], COPD, *Cytomegalovirus*, TB [7], and transplant patients [33]	70% of immunocompromised patients [4]	Triazole resistance [12] and echinocandins [4]
	*Candida albicans*	TB [7]	30–40% [4])	Echinocandins, flucytosine [4]

## Data Availability

No new data was curated in this review, which was based on a critique of existing best-published information shared in journals and books.

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
