# Peer review of "Pathogenic Drug Resistant Fungi: A Review of Mitigation Strategies"

_ijms, 2023, doi:10.3390/ijms24021584_

Round 1
Reviewer 1 Report
This up-to-date and important review on pathogenic fungi provides an insight into challenges and implications on the topic of antifungal drug resistance along with discussing effectiveness of established disease mitigation modalities and approaches. Considering the emerging antifungal crisis, this review has a potential wild interest. The paper is well-written and extensive. Taking into account this and the wild potential interest, the manuscript should be considered for publication after minor revision.
Below, some minor comments and suggestions are listed to improve the quality of the present manuscript.
P1/L23: “Fungi are eukaryotic microbial species present as either yeast, fungi, mould or dimorphic forms.” – The “fungi” between yeast and mould is misleading, because the group of fungi includes microorganisms such as yeasts and molds and mushrooms. Please, rephrase this sentence.
P1/L28-31: Please mention nycorrhizal fungi.
P1/L31-32: Please delete: e.g., Botrytis cinerea
P1/L45: Pneumocystis in Italic
P2/L56: Aspergillus fumigatus in Italic
P2/L57: Sarocladium kiliense in Italic
P2/L97: Candida in Italic
P3/L102: “…the causative agents of disease…” - the most important causative agents of disease
P4/L115: “Dermatophyte species are zoonotic…” – The most of Dermatophyte species are zoonotic
P4/L130-131: “Candida and Aspergillus” - Candida species and Aspergillus fumigatus”
P4/L144: The term invasive fungal infection is used only to characterize systemic, generalized, deep-seated, visceral and severe, life-threatening fungal infections, in contrast to superficial, local, benign, self-limiting fungal diseases. Please, consider the term of invasive fungal infection and rephrase the sentence according to this.
P4/L147-148: C. glabrata- Candida glabrata (Italic). It is the first appearing of this fungal species in the text, therefore do not abbreviate the genus name.
P4/L157: C. neoformans or C. gattii - Candida neoformans or Candida gattii (both in Italic). It is the first appearing of this fungal species in the text, therefore do not abbreviate the genus name.
P5/L162: aspergillus – Aspergillus (Italic)
P5/ 170-172: “Many A. fumigatus isolates are resistant to triazoles and possess pan-azole resistance. A. niger for example has resistance to oral itraconazole and isovuconazole drugs with A. terreus and A. nidulans possessing resistance to AMPB [47].” – Please, use the full species name Aspergillus fumigatus, Aspergillus terreus, Aspergillus nidulans (all in Italic.). It is the first appearing of this fungal species in the text, therefore do not abbreviate the genus name.
P5/L193: C. tropicalis – Candida tropicalis (in Italic) It is the first appearing of this fungal species in the text, therefore do not abbreviate the genus name.
P5/L199: multi antifungal – multi-antifungal
P5/L204: “…in Candida auris and Candida glabrata…” - Please, abbreviate the genus name as C. as C. auris, C. glabrata (both in Italic) because these species already appeared in the text.
P6/L229: “(infections by Candida)” - (infections by Candida spp.), Candida in Italic
P6/L233: Delete the “.” at the end of the sentence.
P6/L241: “(including fungal sp.)” – (including fungal species)
P6/L246: VH2O2 – Please, put both 2 in lower case.
P6/L247: gas (EO) – Please, delete “(EO)” because this abbreviation is not used in the text.
P7/L265: glutaraldehyde – Please, abbreviate glutaraldehyde as GTA.
P7/L277: Bacillus in Italic
P7/L280: “cells” – bacterial cells
P7/L285: “high level disinfection (HLD)“ – Before high level disinfection is already abbreviated, please use jut the HLD abbreviation.
P7/L287: ”…oocysts (Cryptosporidium sp.) or cysts (Giardia sp)…” – Please clarify one or more species, sp. or spp.?
P7/L295: Please, delete the “.” after [69].
P8/L312: What does "GI" abbreviate? Please, indicate it in brackets.
P8/L316: Please, delete the (AER) abbreviation because it is not used in the remaining text.
P8/L335: Please, delete a “.” at the end of the sentence.
P8/L336: C. albicans in Italic
P8/L343-344: Candida in Italic
P8/L347: Candida auris – C. auris (Italic).
P9/L351: “genes” – What kind of genes? Please discuss it shortly.
P9/L359-360: “…silver (AgNPs)7, copper 359 (CuNPs) or titanium (TiO2 NPs)…” - silver (Ag NPs)7, copper (Cu NPs) or titanium dioxide (TiO2 NPs)
P9/L369: MBIC - What does MBIC abbreviate? Please, indicate it.
P9/L372: [8]7 – Please, clarify the reference.
P9/L375: Please, delete “[“.
P9/L384: Candida albicans – C. albicans (Italic)
P9/L391: Please, delete the (ROS) abbreviation because it is not used in the remaining text.
P10/L403 and 408: in vivo and in vitro are in in Italic.
P10/L408: Please delete one of the “.” at the end of the sentence.
P10/L417-418: fungal priority pathogen list - FPPL
P10/L433: antibiotic – antifungal
P10/L435: biofilm-harbored – biofilm-forming
P10/L439: Please delete one of the “.” at the end of the sentence.
P10/L443: noy – not
Table 1: All “spp.” are not Italic, Paracoccidioides spp - Paracoccidioides spp.
Figure 1: Low resolution figure! Please, improve the resolution. ex. = e.g.?
Author Response
Comments:
This up-to-date and important review on pathogenic fungi provides an insight into challenges and implications on the topic of antifungal drug resistance along with discussing effectiveness of established disease mitigation modalities and approaches. Considering the emerging antifungal crisis, this review has a potential wild interest. The paper is well-written and extensive. Taking into account this and the wild potential interest, the manuscript should be considered for publication after minor revision.
Below, some minor comments and suggestions are listed to improve the quality of the present manuscript.
P1/L23: “Fungi are eukaryotic microbial species present as either yeast, fungi, mould or dimorphic forms.” – The “fungi” between yeast and mould is misleading, because the group of fungi includes microorganisms such as yeasts and molds and mushrooms. Please, rephrase this sentence.
Response:
Sentence adjusted.
Comments:
P1/L28-31: Please mention nycorrhizal fungi.
Response:
Added to text.
P1/L31-32: Please delete: e.g., Botrytis cinerea
Response:
Deleted.
P1/L45: Pneumocystis in Italic
P2/L56: Aspergillus fumigatus in Italic
P2/L57: Sarocladium kiliense in Italic
P2/L97: Candida in Italic
P3/L102: “…the causative agents of disease…” - the most important causative agents of disease
P4/L115: “Dermatophyte species are zoonotic…” – The most of Dermatophyte species are zoonotic
P4/L130-131: “Candida and Aspergillus” - Candida species and Aspergillus fumigatus”
P4/L144: The term invasive fungal infection is used only to characterize systemic, generalized, deep-seated, visceral and severe, life-threatening fungal infections, in contrast to superficial, local, benign, self-limiting fungal diseases. Please, consider the term of invasive fungal infection and rephrase the sentence according to this.
Response:
Author does not see this sentence in the text and is unsure what the comment means.
P4/L147-148: C. glabrata- Candida glabrata (Italic). It is the first appearing of this fungal species in the text, therefore do not abbreviate the genus name.
P4/L157: C. neoformans or C. gattii - Candida neoformans or Candida gattii (both in Italic). It is the first appearing of this fungal species in the text, therefore do not abbreviate the genus name.
P5/L162: aspergillus – Aspergillus (Italic)
P5/ 170-172: “Many A. fumigatus isolates are resistant to triazoles and possess pan-azole resistance. A. niger for example has resistance to oral itraconazole and isovuconazole drugs with A. terreus and A. nidulans possessing resistance to AMPB [47].” – Please, use the full species name Aspergillus fumigatus, Aspergillus terreus, Aspergillus nidulans (all in Italic.). It is the first appearing of this fungal species in the text, therefore do not abbreviate the genus name.
P5/L193: C. tropicalis – Candida tropicalis (in Italic) It is the first appearing of this fungal species in the text, therefore do not abbreviate the genus name.
P5/L199: multi antifungal – multi-antifungal
P5/L204: “…in Candida auris and Candida glabrata…” - Please, abbreviate the genus name as C. as C. auris, C. glabrata (both in Italic) because these species already appeared in the text.
P6/L229: “(infections by Candida)” - (infections by Candida spp.), Candida in Italic
P6/L233: Delete the “.” at the end of the sentence.
P6/L241: “(including fungal sp.)” – (including fungal species)
P6/L246: VH2O2 – Please, put both 2 in lower case.
P6/L247: gas (EO) – Please, delete “(EO)” because this abbreviation is not used in the text.
P7/L265: glutaraldehyde – Please, abbreviate glutaraldehyde as GTA.
P7/L277: Bacillus in Italic
P7/L280: “cells” – bacterial cells
P7/L285: “high level disinfection (HLD)“ – Before high level disinfection is already abbreviated, please use jut the HLD abbreviation.
P7/L287: ”…oocysts (Cryptosporidium sp.) or cysts (Giardia sp)…” – Please clarify one or more species, sp. or spp.?
P7/L295: Please, delete the “.” after [69].
P8/L312: What does "GI" abbreviate? Please, indicate it in brackets.
P8/L316: Please, delete the (AER) abbreviation because it is not used in the remaining text.
P8/L335: Please, delete a “.” at the end of the sentence.
P8/L336: C. albicans in Italic
P8/L343-344: Candida in Italic
P8/L347: Candida auris – C. auris (Italic).
P9/L359-360: “…silver (AgNPs)7, copper 359 (CuNPs) or titanium (TiO2 NPs)…” - silver (Ag NPs)7, copper (Cu NPs) or titanium dioxide (TiO2 NPs)
P9/L369: MBIC - What does MBIC abbreviate? Please, indicate it.
P9/L372: [8]7 – Please, clarify the reference.
P9/L375: Please, delete “[“.
P9/L384: Candida albicans – C. albicans (Italic)
P9/L391: Please, delete the (ROS) abbreviation because it is not used in the remaining text.
P10/L403 and 408: in vivo and in vitro are in in Italic.
P10/L408: Please delete one of the “.” at the end of the sentence.
P10/L417-418: fungal priority pathogen list - FPPL
P10/L433: antibiotic – antifungal
P10/L435: biofilm-harbored – biofilm-forming
P10/L439: Please delete one of the “.” at the end of the sentence.
P10/L443: noy – not
Table 1: All “spp.” are not Italic, Paracoccidioides spp - Paracoccidioides spp.
Response:
All the typos, grammatical errors and adjustments listed above have been corrected.
Comment:
Figure 1: Low resolution figure! Please, improve the resolution. ex. = e.g.?
Response:
Figure 1 has been adjusted

Reviewer 2 Report
I suggested authors could consider to add some comments of mycotoxin produced by fungus-induced toxicity and health risk.
Author Response
Reviewer additional comment - I suggested authors could consider to add some comments of mycotoxin produced by fungus-induced toxicity and health risk.
Author response - agreed, this is a helpful point. We have included a sub-section on this (6) given its relevance. We feel that we have captured the main issues in terms of mycotoxins, toxicity and health risks in the context of decontamination challenges (this paper). Its an area that the last author has worked with closely since his postdoctoral days with Prof John E Smith at Strathclyde University (early 90s) that remains very important today. Also the addition of 13 appropriate references. We also includes a comment in the summary and abstract to ensure coverage. We are mindful that this sub-topic is considerable verbose, thus, we kept it succinct and informative.
Thank you.

Reviewer 3 Report
This review provides a timely insight into challenges and implications on the topic of antifungal drug resistance along with discussing effectiveness of established disease mitigation modalities and approaches. This work is meaningful and I think it can be published. Meanwhile, some issues need to be improved before publication in the following:
Lines 17-21, these sentences can not reflect the innovation of the article and need to be rewritten.
Some references lack article numbers, and some journals do not use abbreviations. 5, 11, 17 et al.
There is drug resistance in the title of the article, but there is less discussion in the text. It is better to discuss drug resistance of pathogenic bacteria. The discussion on the mechanism of drug resistance should be added to provide guidance for improving drug resistance and creating new drugs.
It is better to provide the chemical structure formula of related drugs for the convenience of readers.
Author Response
Ijms-2168706
Reviewer #2
This review provides a timely insight into challenges and implications on the topic of antifungal drug resistance along with discussing effectiveness of established disease mitigation modalities and approaches. This work is meaningful and I think it can be published. Meanwhile, some issues need to be improved before publication in the following:
Author response : we thank Reviewer #2 for very positive comments on our timely paper. This also strongly echoes Reviewer 1 disposition.
Lines 17-21, these sentences can not reflect the innovation of the article and need to be rewritten.
Author response – agreed, this was changed as requested.
Some references lack article numbers, and some journals do not use abbreviations. 5, 11, 17 et al.
Author response: agreed, this was addressed – we also took opportunity to apply italics to fungal species etc that we had missed.
There is drug resistance in the title of the article, but there is less discussion in the text. It is better to discuss drug resistance of pathogenic bacteria. The discussion on the mechanism of drug resistance should be added to provide guidance for improving drug resistance and creating new drugs. It is better to provide the chemical structure formula of related drugs for the convenience of readers.
Author response:
Addressed – title has been amended – we have avoided discussing drug resistance of bacteria as the focus in on fungi. We have provide a sentence to link to chemical formula for related antifungal drugs (line 71-72), which was a very helpful suggestion.
